# Biotechnological Potential of the Stress Response and Plant Cell Death Regulators Proteins in the Biofuel Industry

**DOI:** 10.3390/cells12162018

**Published:** 2023-08-08

**Authors:** Maciej Jerzy Bernacki, Jakub Mielecki, Andrzej Antczak, Michał Drożdżek, Damian Witoń, Joanna Dąbrowska-Bronk, Piotr Gawroński, Paweł Burdiak, Monika Marchwicka, Anna Rusaczonek, Katarzyna Dąbkowska-Susfał, Wacław Roman Strobel, Ewa J. Mellerowicz, Janusz Zawadzki, Magdalena Szechyńska-Hebda, Stanisław Karpiński

**Affiliations:** 1Department of Plant Genetics, Breeding and Biotechnology, Institute of Biology, Warsaw University of Life Sciences, Nowoursynowska Street 159, 02-776 Warsaw, Poland; maciej_bernacki@sggw.edu.pl (M.J.B.); jakub_mielecki@sggw.edu.pl (J.M.); damian_witon@sggw.edu.pl (D.W.); piotr_gawronski1@sggw.edu.pl (P.G.); pawel_burdiak@sggw.edu.pl (P.B.); 2Institute of Technology and Life Sciences—National Research Institute, Falenty, Al. Hrabska 3, 05-090 Raszyn, Poland; w.strobel@itp.edu.pl; 3Institute of Wood Sciences and Furniture, Warsaw University of Life Sciences—SGGW, 02-776 Warsaw, Poland; andrzej_antczak@sggw.edu.pl (A.A.); michal_drozdzek@sggw.edu.pl (M.D.); monika_marchwicka@sggw.edu.pl (M.M.); janusz_zawadzki@sggw.edu.pl (J.Z.); 4Department of Plant Physiology, Institute of Biology, Warsaw University of Life Sciences, 02-776 Warsaw, Poland; joanna_dabrowska@sggw.edu.pl; 5Department of Botany, Institute of Biology, Warsaw University of Life Sciences, 02-776 Warsaw, Poland; anna_rusaczonek@sggw.edu.pl; 6Faculty of Chemical and Process Engineering, Warsaw University of Technology, 00-645 Warsaw, Poland; katarzyna.susfal@pw.edu.pl; 7Department of Forest Genetics and Plant Physiology, Umeå Plant Science Centre, Swedish University of Agricultural Sciences, 901-83 Umeå, Sweden; ewa.mellerowicz@slu.se; 8W. Szafer Institute of Botany Polish Academy of Sciences, Lubicz 46, 31-512 Kraków, Poland; m.szechynska-hebda@botany.pl

**Keywords:** poplar, lignin, bioethanol, cell wall, scarification, cell death, biofuels

## Abstract

Production of biofuel from lignocellulosic biomass is relatively low due to the limited knowledge about natural cell wall loosening and cellulolytic processes in plants. Industrial separation of cellulose fiber mass from lignin, its saccharification and alcoholic fermentation is still cost-ineffective and environmentally unfriendly. Assuming that the green transformation is inevitable and that new sources of raw materials for biofuels are needed, we decided to study cell death—a natural process occurring in plants in the context of reducing the recalcitrance of lignocellulose for the production of second-generation bioethanol. “Members of the enzyme families responsible for lysigenous aerenchyma formation were identified during the root hypoxia stress in *Arabidopsis thaliana* cell death mutants. The cell death regulatory genes, LESION SIMULATING DISEASE 1 (LSD1), PHYTOALEXIN DEFICIENT 4 (PAD4) and ENHANCED DISEASE SUSCEPTIBILITY 1 (EDS1) conditionally regulate the cell wall when suppressed in transgenic aspen. During four years of growth in the field, the following effects were observed: lignin content was reduced, the cellulose fiber polymerization degree increased and the growth itself was unaffected. The wood of transgenic trees was more efficient as a substrate for saccharification, alcoholic fermentation and bioethanol production. The presented results may trigger the development of novel biotechnologies in the biofuel industry.

## 1. Introduction

One of the most important challenges in the 21st century is to provide reliable, renewable and environmentally friendly sources of energy. Moreover, the current geopolitical situation forces the invention of an alternative solution to traditional energy sources even more urgently. One of the answers is improvement of renewable biofuels production. Currently, bioethanol is still mostly produced as a first-generation biofuel from edible plants. As a result, it reduces food sources and thus is questionable. Although lignocellulosic biomass is widely obtained from tree plantations or agricultural waste, only ~3% of bioethanol is produced from this source as second-generation bioethanol [1,2,3,4]. The main limitation of efficient, industrial production of second-generation bioethanol is its complex composition and the structure of plant cell walls. Mostly composed of cellulose, the cell wall is highly heterogeneous as it also contains hemicelluloses, pectin and lignin. The outcome structure is determined by the function of secondary cell walls [5,6,7]. During technological processing, lignocellulosic biomass requires the separation of its components and breaking down complex polysaccharides into fermentable sugars [8,9]. In particular, lignin presents a great challenge as it affects the activity of cellulases [9,10] and also because phenolic compounds are inhibitors of alcohol fermentation in brewers’ yeast (*Saccharomyces cerevisiae*) [11]. Pretreatments to reduce recalcitrance increase the cost of production and negatively impact the environment with toxic wastes [12].

Plants are capable of efficient saccharification in vivo during induction of certain types of Programmed Cell Death (PCD) [13] e.g., during lysigenous aerenchyma formation in response to root hypoxia [14]. Discovered proteins witch regulated lysigenous aerenchyma formation opened a new pathway to study this form of plants adaptation to stress. We also postulated that the ability of plants to degradation of their cell wall can be an interesting target to improve second-generation bioethanol production. According to our knowledge, plant enzymes involved in the natural saccharification of the cell wall have not been explored in the context of their use as a potential source for industrial enzymes. Similarly, the role of genes encoding molecular PCD regulators has not been studied in the cell wall lignification and cellulose fiber polymerization-related processes.

For this study, we used two model plants in biotechnology, well-established *Arabidopsis thaliana* and *Populus tremula x tremuloides* which have been proposed as the best tree for genetic research and considered as *Arabidopsis* for forestry [15]. Although Arabidopsis thaliana is a herbaceous and annual plant, it shows the ability to form secondary xylem. It was shown that when *Arabidopsis thaliana* plants grown in appropriate conditions exhibit secondary growth in the hypocotyl, with the development of both a vascular and a cork cambium. *Arabidopsis thaliana* was previously proposed as a model for wood formation in trees [16]. We believe that our approach is valuable because it shows that the role of the tested proteins is strongly conserved in the plant kingdom, which makes it possible to transfer the knowledge obtained to other crop species.

Here we present the biotechnological potential of the conditional PCD regulators [17,18,19] i.e., LESION SIMULATING DISEASE 1 (LSD1), ENHANCED DISEASE SUSCEPTIBILITY 1 (EDS1), and PHYTOALEXIN DEFFICIENT 4 (PAD4). The very regulators were earlier recognized as involved in the lysigenous aerenchyma formation in response to root hypoxia stress in *Arabidopsis thaliana* [14]. Conditional PCD regulation by LSD1, EDS1 and PAD4 was demonstrated in runaway PCD during the growth of *lsd1* mutant in ambient laboratory conditions and in field conditions [17,18], which affected its seed yield and productivity. Based on the results in stable transgenic aspen lines with deregulated PtEDS1, PtLSD1, and PtPAD4, we obtained four independent transgenic lines with lower lignin content and a higher cellulose polymerization degree in four-year-old aspen wood. Higher efficiency of the industrial bioethanol production from transgenic wood with the lowest lignin content was found.

## 2. Materials and Methods

### 2.1. Arabidopsis thaliana Plant Material

*Arabidopsis thaliana* seeds used in this study were available in our lab. Experiments were performed using *lsd1*, *eds1*, *lsd1/eds1* mutants in two ecotypes, Col-0 and Ws-0. 

### 2.2. Populus tremula × tremuloides Plant Material and Genetic Transformation 

The silencing constructs for *LSD1*/*PAD4* (XM_002317030, XM_002310499) and *EDS1*/*PAD4* (XM_002318590, XM_002310499) genes were carried out based on the RNA interference using a binary vector pH7GWIWG2(I) under the control of the 35S promoter [20] (Appendix A). Short cDNA sequences of target genes were amplified using primers described in the Appendix A. Inserts of analyzed genes were introduced into the pH7GWIWG2(I) vector according to the ‘Gateway’ technology [20] and transferred to *Agrobacterium tumefaciens* C58. Next, some bacteria were used in the transformation of hybrid aspen plants. Transgenic hybrid aspen lines (*P. tremula × tremuloides*, background T89) were generated at the Umeå Plant Science Center according to the transformation procedure [21] from among 19 independent transgenic *P. tremula × tremuloides* lines, 4 of them with significant deregulation in expression levels of target genes were chosen for further experiments. For it is known that in *Arabidopsis thaliana LSD1*, *EDS1* and *PAD4* are co-regulated, we checked the expression level of *PtLSD1*, *PtEDS1* and *PtPAD4* in aspen leaves from field conditions. Based on these results we described the following lines: Lines 1, 2, 3 and 4.

### 2.3. Aspen Growth Condition 

Transgenic aspen lines were propagated under in vitro conditions. Plants were grown in phytotron during a long photoperiod (16 h/8 h day/night) with a light intensity of 120 μmol photons m^−2^ s^−1^, temperature 25 °C/20 °C day/night and relative humidity 55%. Every 2 weeks, plants were transferred to a fresh Murashige and Skoog (MS) medium [22]. Well-rooted plants were transferred to the greenhouse, kept 3 days in open jars to acclimatize and then transferred to pots with soil. Transgenic and controlled aspen plants were grown in the greenhouse under natural light supported by low-pressure sodium lamps (Philips, Amsterdam, Netherlands) emitting light intensity (300 μmol photons m^−2^ s^−1^). After 9 months the transgenic and control lines were transplanted to a field and grown in the Wolica experimental field (52°8′30″ N, 21°4′12″ E) in 2017. Experiments and measurements were performed on 4-year-old transgenic and wild-type aspen plants. Trees were randomized in the experimental field to minimize the effect of placement on observed phenotype. The scheme of the field experiment is shown in the figure (Appendix A). We performed analysis on all four independent transgenic lines (with a minimum of six trees *per* Line) obtained from independent transformation events as we wanted to be sure that the phenotype that we observe is a consequence of targeted genetic manipulation and not a side effect of transgene integration into aspen genome.

### 2.4. Inducing of Lysigenous Aerenchyma Formation by Root Hypoxia Stress

In general, the experiment was conducted as described [14] above. *Arabidopsis thaliana* was grown in short-day photoperiod conditions (8 h of day and 16 h of night) for 8 weeks. After that, they were submerged in degassed water. The oxygen content in the water was reduced by boiling. After it cooled down, we checked the O_2_ content in water using an oxygen sensor DO210 (Extech Instruments, Nashua, NH, USA) before submerging the plants. An eventual evaporation was mitigated by topping up the water. After 7 days of hypoxia stress, hypocotyls and main roots material were harvested for future analysis (Appendix A).

### 2.5. Determining to Reduce Sugar Content in Arabidopsis thaliana Hypocotyls

Reducing sugar content was determined using the Amplex Red Glucose Assay kit (Thermo Fisher Scientific, Waltham, MA, USA) accordingly to the manufacturer’s protocol. 

### 2.6. Cellulase and Xylanase Activity Assays

Proteins from hypocotyls were extracted using a plastic homogenizer pestle and 1.5 mL eppendorf type tube in the PBS buffer on the ice at 4 °C. Protein content in samples was measured using the Bradford method. For activity analysis we used 20 μg of total protein. Cellulase and xylanase activities were measured against CMC (Sigma Aldrich, St. Louis, MO, USA) and beechwood xylan (Roth, Karlsruhe, Deutschland) as substrates, both at 1% (W:V) concentration. We measured the reducing sugar content in each sample before adding substrate and normalized results so that they could reflect how much the reducing sugars were produced from the substrate during the assay and not during the stress itself. We measured reducing sugar content after every hour, enzymatic reactions were carried out up to 3 h. 

### 2.7. Native-PAGE Separation and Zymographic Activity Assay

We separated 20 ug of total proteins from each sample on 8% polyacrylamide gel. As we wanted to visualize the enzymatic activity, we did not use any reducing or denaturizing agents such as SDS, β-mercaptoethanol or DTT. Gels were supplemented with 1% CMC as substrate. After separation (2 h, 60–70 V) at 4 °C each gel was submerged in PBS and incubated at room temperature for 18 h. The gels were stained with Congo Red (Merck, Darmstadt, Germany) which binds to glycosidic bonds in cellulose. Fifteen min later the gels were detained using 2 M NaCl until clear zones were observed on the gel which indicate activity of cellulases (Appendix A).

### 2.8. Maldi-TOF Analysis of Isolated Proteins

Bands exhibiting cellulase activity were cut off from each sample and sent to analysis on Maldi-TOF to The Environmental Mass Spectrometry Laboratory in the Department of Biophysics of the IBB PAN (Warsaw, Poland). Samples were treated with trypsin before separating and analyzing them on a mass spectrometer. The results were analyzed using the Mascot software and trimmed so there would be no more than 1% of false positives in each sample. The equipment used was sponsored in part by the Centre for Preclinical Research and Technology (CePT), a project co-sponsored by the European Regional Development Fund and Innovative Economy, The National Cohesion Strategy of Poland.

### 2.9. RNA Isolation and qPCR Analysis

RNA isolation, cDNA synthesis and qPCR analysis were performed as described before [23]. UBQ (LP-GCTTGAAGATGGGAGGACTCT, RP-AAATCTGCATTCCTCCACGG), PP2A (LP-ACTCTCTGCACTGTTGAGGAA, RP-ACCAATCAACCAGGTCAGTCT) and ACT (LP-TTACATGTTCACCACCACTGC RP-CTGCTCATAGTCAAGGGCAAC) were used as reference genes. For checking LSD1, EDS1 and PAD4 the following primers were used: LSD1 (LP-GACTCAAACTGTCGTTGTGGA, RP-CTGTAGTGACACCAACAACAAC, EDS1 (LP-CGACACCTCAAGAATGAAGA, RP-GTGCATTTATACCGTCTTGG), PAD4 (LP-CATCACCTTTGGCTCTCCAT, RP-AAATTTGCCACCCCATCTTT).

### 2.10. RNAseq Analysis

Fastq files were trimmed using TrimGalore 0.6.6 (https://github.com/FelixKrueger/TrimGalore (accessed on 20 June 2021)) with the following settings: --paired -j 4 --fastqc --quality 30 --length 50 --clip_R1 10 --clipR2 10 --three_prime_clip_R1 10 --three_prime_clip_R2 10. The quality of trimmed reads was assessed using the FastQC version (https://github.com/s-andrews/FastQC (accessed on 20 June 2021)). Reads were mapped to the Populus trichocarpa v3.0 genome [24] with annotations in version 4.1 downloaded from JGI (https://data.jgi.doe.gov/ (accessed on 20 June 2021)) using STAR aligner 2.7.6a [25] with the following settings: --outFilterScoreMinOverLread 0.3 --outFilterMatchNminOverLread 0.3 --outFilterMultimapNmax 100 --outReadsUnmapped Fastx --alignIntronMax 11,000 --quantMode GeneCounts. After mapping reads were counted using HTseq with the settings: -f bam -r pos -s no. Differentially expressed genes were identified with DEseq2 [26].

### 2.11. Biometric Parameters and Gas Exchanging Measurement

Trees were weighed and measured using an industrial scale and a commercially available measure. CO_2_ measurement was performed using LI-6400XT Portable Photosynthesis System (Li-COR Inc., Lincoln, NE, USA) just as described before [23]. 

### 2.12. Determination of Photosynthetic Pigments Content

The photosynthetic pigments measurements were determined as precisely described in the previous chapters [27]. Pigments extracted from the frozen tissue (grinded in liquid nitrogen before and stored at −80 °C) were separated on a SynergiTM 4 μm Max-RP 80Å 250 × 4.6 mm column (Phenomenex, Torrance, CA, USA) at 30 °C using Shimadzu HPLC System (Shimadzu, Kyoto, Japan). The results were expressed as peak area per μg of fresh weight.

### 2.13. Gene Ontology

Gene ontology analyses were performed using Panther (http://www.pantherdb.org/ (5 August 2022).

### 2.14. Cellulose, Hemicellulose and Lignin Content

Dry wood material was milled using a Retsch mill (SM 200, Retsch, Germany) and sieved (AS 200, Retsch, Germany). Wood powder with particle sizes from 0.43 mm to 1.02 mm was used for chemical analyses. The material was extracted in Soxhlet apparatus with a mixture of chloroform-ethanol 93:7 *w*/*w* for 10 h before analyses. Cellulose content was determined by the Kürschner–Hoffer method and holocellulose content was analyzed according to Wise [28]. Mineral substances content was determined by [29] on dust fraction (under 0.43 mm) and lignin content was determined according to TAPPI UM 250 (1985) and TAPPI T222 om-02 (2006). 

### 2.15. The Alkaline Pre-Treatment and Enzymatic Hydrolysis of Aspen Wood

For the alkaline pre-treatment, a 2% aqueous NaOH solution was used in the proportions of 100 cm^3^ of sodium hydroxide per 5 g of dry aspen wood powder from the fraction of 0.43 mm to 1.02 mm. The alkaline pre-treatment was performed in an autoclave under the following conditions: 121 °C, *p* = 0.1 MPa, 30 min. Afterward the activated material was subjected to enzymatic hydrolysis according to the procedure described by us previously [30] with some changes specified below. The hydrolysis Cellic CTec2 enzyme (Novozymes, Krogshoejvej, Denmark) was used and activity was 148 FPU/cm^3^ determined by the NREL method [31]. The enzymatic hydrolysis process was carried out in the Erlenmeyer flasks and the total volume of the mixture was 100 cm^3^. In the process 50 cm^3^ of 0.1 M citrate buffer solution at pH = 4. was used and due to the use of yeast in the next step, no sodium azide was added during the hydrolysis. In the hydrolysis process, 3.325 cm^3^ of 25% (*v*/*v*) Cellic CTec2 enzyme solution on each sample was used in the ratio of 1 g of concentrated enzyme to 1 g of absolutely dry biomass. The hydrolysis proceeded in an incubator IKA KS 3000i control (IKA, Warsaw, Poland) with shaking of 150 rpm at 50 °C. The total time of enzymatic hydrolysis was 72 h. After the very process, glucose and xylose concentrations in the supernatant were analyzed by the HPLC method.

### 2.16. The Ethanol Fermentation Process

*Saccharomyces cerevisiae* KKP 50 yeast strain on YPD agar plate obtained from the Institute of Agricultural and Food Biotechnology culture collection (Warsaw, Poland) was used in the fermentation study. Yeast inoculum was prepared by transferring single colonies from the YPD agar plate using a sterile loop into 25 cm^3^ of autoclaved (15 min at 121 °C) liquid YPD medium (Millipore, Burlington, MA, USA) in 100 cm^3^ Erlenmeyer flask and incubating flask overnight at 30 °C with shaking of 240 rpm. Fermentation broths were prepared as follows: 20 cm^3^ of each enzymatic hydrolysate in 100 cm^3^ Erlenmeyer flasks were supplemented with 0.2 g of yeast extract (Biocorp, Issoire, France) and autoclaved for 15 min at 121 °C. Subsequently, 1 cm^3^ of inoculum cultured overnight was added into each flask. Batch cultures were carried out in triplicate at 30 °C with shaking of 240 rpm for 47.5 h. During cultivation 0.5 cm^3^ samples were taken from fermentation broths to 1.5 cm^3^ Eppendorf tubes, cooled in ice water and then centrifuged for 10 min.) to remove the yeast biomass. The obtained supernatants were frozen until the HPLC analysis. 

### 2.17. The HPLC Analysis

The concentration of glucose and xylose in the enzymatic hydrolysates (before fermentation) or glucose, xylose and ethanol in the fermentation broths (after fermentation) were determined using the high-performance liquid chromatography (HPLC, Varian 635 CL System, USA) with a refractive index (RI) detector (Smartline 2300, Knauer, Germany) and thermostated at 60 °C Rezex ROA Organic acid H+ (8%) (300 × 7.8 mm) column with Security Guard Cartridge Carbo-H (4 × 3.0 mm) (Phenomenex, Torrance, CA, USA) using 0.001 N H_2_SO_4_ as the mobile phase at a flow rate of 0.4 cm^3^ min^−1^. All the HPLC analyses were executed in triplicate. The concentrations were expressed in g * dm^−3^ of hydrolysates or fermentation broths.

### 2.18. The Analysis of Cellulose Polymerization Degree

The degree of cellulose polymerization was determined using the SEC (Size Exclusion Chromatography) technique. Cellulose was isolated from aspen wood with the Kürschner-Hoffer method, dissolved in 8% lithium chloride/N, N-dimethylacetamide system and analyzed as described in detail in the previous publications [8,15]. All of the SEC analyses were executed in triplicate or six times.

### 2.19. Statistic

For the aspen wood properties, biometric measurement and *Arabidopsis thaliana* the statistical analysis was performed using the Tukey honest significant difference (HSD) test. The number of biological and technical repetitions can be found in the description of the Figures. The qPCR data statistical analysis was performed using LinReg.

## 3. Results

### 3.1. EDS1 and LSD1 Are Involved in Lysigenous Aerenchyma Formation

It was confirmed that LSD1, EDS1 and PAD4 are involved in the lysigenous aerenchyma formation [14] (Figure 1 and Appendix A). Changes in the content of reducing sugars in roots and hypocotyls were determined in plants growing in controlled conditions and in plants flooded for 7 days (root hypoxia stress, see material and methods) to establish a cell wall saccharification and lysis occurring during the formation of lysigenous aerenchyma in hypocotyls. The elevated level of reducing sugars before hypoxia was only observed in the *lsd1* mutant when compared to the wild-type (Ws-0) plant (Figure 1A), which is related to the fact that this mutant induces cell death in long day conditions without the action of external stress factors (Appendix A) [32]. After the stress, the *lsd1* mutant exhibited an elevated (compared to Ws-0) but stable (as compared to *lsd1* plants before hypoxia) level of reducing sugars. It is related to the fact that *lsd1* exhibit enhanced cell death levels in long-day conditions in comparison to the wild type [32]. In contrast, the *eds1* mutant exhibited the highest level of reducing sugars, whereas the *eds1/lsd1* double mutant did not differ from Ws-0 after 7 days of stress (Figure 1B) Moreover we found no difference between WT and double *lsd1/eds1* mutant in both conditions (Figure 1A,B). This is caused by the actions of these two proteins which are closely related and LSD1 is antagonistic to EDS1. It was many times showed that phenotype of *lsd1/eds1* is similar to the WT phenotype, for example in response to various types of biotic and abiotic stresses [17,18]. These results suggest that both LSD1 and EDS1 are regulators of saccharification and lysis occurring during the formation of lysigenous aerenchyma in hypocotyls. However, LSD1 seems to be a regulator itself, while EDS1 acts as a regulator together with PAD4, and the regulatory role of EDS1-PAD4 [33] in this process is important under stress conditions.

### 3.2. In Vitro Cellulolytic and Xylanolytic Activities of Enzymes Isolated from eds1 Hypocotyls

Two different substrates i.e., carboxymethyl cellulose (CMC) and beechwood xylan were added to proteins extracted from Ws-0 and *eds1* hypocotyls. CMC is a substrate for β-D-glucanase hydrolyzing β-1-4 glycosidic linkages between glucopyranose units (in amorphous cellulose), while xylan is a substrate for xylanase acting on β-1-4 glycosidic linkages between xylopyranose units in xylan backbone. The *Arabidopsis thaliana* enzymes induced in roots and hypocotyls during hypoxia stress have the potential to break down both cellulose and xylan, however, proteins extracted from the *eds1* mutant exhibited higher activity than those extracted from Ws-0 (Figure 1C,D). In contrast, enzymatic extracts from *lsd1* and *lsd1*/*eds1* double mutant exhibited lower activity against both CMC and xylan than *eds1* mutant and wild-type plants (Appendix A). The high reducing sugar content after the stress and stable enzymatic activity of extracts of *eds1* mutant indicated that cellulase, xylanases and also cell wall loosening (CWL) proteins were intact and unaffected probably by endogenous proteolysis. Similarly, higher activity was observed in the protein extracts from *eds1* in the Col-0 background examined in the zymography activity assay (Appendix A; proteins were separated with Native-PAGE in the presence of 1% CMC as substrate). The ecotype Col-0 showed lower activity than Ws-0 (Appendix A) and, in the same way, it was observed in the *eds1* mutant. Generally, proteins with molecular weight higher than 150 kDa had enzymatic activity.

### 3.3. Enzymes Generating Reducing Ends Were Found in Hypocotyls and Roots Undergoing Lysigenous Aerenchyma Formation

In the protein fraction with enzymatic activity that was isolated from the zymographic activity assay, many proteins in individual genotypes were identified during the Maldi-TOF analysis (Appendix A). Some of them were found in two or more genotypes, but after removing duplicates, we found a list of 563 proteins (Appendix A). Approximately 6.4% (37 proteins) of all identified proteins were assigned/aligned to known or predicted proteins capable of generating reducing sugars (Figure 2A,B and Appendix A). We identified enzymes from different CAZY families, with known or predicted CWL activities (homogalacturonan, cellulose, xylan, mannan, callose, xyloglucan, and hydroxyproline-rich or arabinogalactan, Appendix A). The protein ontology analysis (Figure 2C and Appendix A) using Panther (Pantherdb.org) software was performed for *eds1* mutants in both ecotypes, due to exhibited saccharification potential. We found four proteins whose abundance depends on EDS1 dysfunction (gene expression inhibition) (Figure 2D). The very proteins likely representing cell wall dismantling machinery could become interesting targets as plant enzymes for future use in plant biotechnology. Most importantly, we found proteins involved in the polysaccharide catabolite processes and proteins involved in the lignin precursor catabolic processes (Appendix A). However, many (120 out of 563) of identified proteins, which could be valuable in biofuels or pulp industry, did not annotate protein classes or other gene ontology classes in the Panther *Arabidopsis thaliana* database (Appendix A). These results confirm that EDS1 is a negative regulator of CWL and saccharification during the formation of lysigenous aerenchyma.

### 3.4. Cell Walls of Aspen with Deregulated PCD Growing in the Field Contain Less Lignin and Longer Cellulose Fibers

It was demonstrated earlier that transgenic aspen with an individually lowered expression of *PtLSD1, PtEDS1,* or *PtPAD4*, differentially modified wood density, wood swelling and cell wall thickness [34,35] in comparison to the wild type in the field conditions. Here four transgenic lines with reduced expression of *PtLSD1, PtEDS1, and PtPAD4* (lines: Line 1, Line 2, Line 3 and Line 4) (Figure 3A–C) were tested. All three genes, i.e., *PtLSD1, PtEDS1, and PtPAD4* were deregulated regardless of the double transgene combination (even if the third gene was not modified in a double transgenic line), probably because these genes are conditionally co-regulated, while LSD1 and PAD4 interact with EDS1 protein [17,18,19,34,35] (Figure 3A–C). Transgenic trees exhibited significantly lowered lignin content in the wood and higher cellulose fiber polymerization degree (Figure 3D). This phenotype (feature) correlated with the expression level of *LSD1*, *EDS1,* and *PAD4* (Appendix A). The relative reduction of lignin (Wt = 100%) was in the range of 10% in Line 1 to 22% in Line 4 (Figure 3D). The cellulose content was also significantly lower (up to 7% in Line 4), however, its polymerization degree was significantly increased to 23% (Figure 3D).

### 3.5. Strong Reduction in Lignin and Higher Cellulose Polymerization Degree in the Wood of Line 4 Allows High Fermentation Efficiency

Lignin is an inhibitor of alcoholic fermentation [36]. Therefore, we confirm that Line 4 was the most suitable for bioethanol production. Higher glucose yield after alkaline pre-treatment and enzymatic hydrolysis of aspen wood (wild-type plants and Line 4) (Figure 3E) as well as higher ethanol yield after fermentation (Figure 3F) were obtained. The high effectiveness of the alcohol fermentation process was confirmed, as almost all the glucose obtained from enzymatic hydrolysis was converted to ethanol for Line 4 wood but not for the wild type (Figure 3G).

### 3.6. PtLSD1, PtEDS1, and PtPAD4 Strongly Influence Cambium and Xylem Transcriptome in Line 4

Having analyzed the transcriptome of cambium and differentiating xylem tissue in Line 4 we found that it differed from the wild-type tree (Appendix A) with 2648 differentially expressed genes (DEG) (Appendix A) of which 2299 were known (recognized by Panther.org, Appendix A) and 342 were unknown (Appendix A). For all the known genes, gene ontology analysis was performed considering their molecular function (Appendix A) or biological process (Appendix A). Many differently expressed genes encoded with putative enzymes (Appendix A). The overrepresentation of genes assigned to the regulation of the phenylpropanoid metabolic process, which is related to lignin biosynthesis [37], was found (Appendix A). Some of the 63 genes (ca. 3.2%) were involved in lignin and cellulose synthesis/metabolism, cell wall biogenesis and cell wall organization (Appendix A). Moreover, we compared genes encoding enzymes identified in the Maldi-TOF experiment in *Arabidopsis thaliana* (Appendix A) with their homologs among genes deregulated in transgenic Line 4. For 19 Line 4 DEGs, its *Arabidopsis* homologs were identified in the Maldi-TOF experiment (Appendix A). This list included two cell wall-loosening proteins and enzymes generating reducing sugars; ALPHA-L-FUCOSIDASE 2 and BETA-D-XYLOSIDASE 7, which could be interesting targets for the biofuel industry. In order to find some gene clusters, which can be particularly important in the context of aspen wood quality, we compared Line 4 DEGs with the known CAZY family protein database (Appendix A) [38] to genes involved in the cell wall integrity (malectin and malectin-like domain-containing proteins (Appendix A) [39]. In Line 4 DEGs 252 we found genes encoding proteins from CAZY families and 23 genes encoding malectin and malectin-like domain-containing proteins. 14 genes from Line 4 DEGs were found in both groups (Appendix A). These results strongly indicate that conditional PCD regulators LSD1, EDS1 and PAD4 are developmental regulators of the secondary cell wall structure in the woody plants.

### 3.7. Transgenic Lines Exhibited Similar Phenotype and Development Level as Wild Type Trees

Insignificant differences in tree height, stem fresh weight and stem diameter during annual growth were measured. In contrast, CO_2_ assimilation was changed in transgenic lines, even if differences were not found in chlorophyll content (Appendix A).

## 4. Discussion

Cell wall biotechnological amelioration with decreased lignin content in the wood a goal of many studies [3,40]. However, the knowledge about cell wall structure and composition regulators, plant cellulolytic and CWL enzymes naturally decomposing cell walls is limited [41,42]. Wood tissue and cell wall development require the occurrence of PCD [43]. Therefore, conditional regulators of PCD and lysigenous aerenchyma formation i.e., LSD1, EDS1, and PAD4 [17,18,19,44,45] were studied in both *Arabidopsis thalian* and aspen for their biotechnological potential in biofuels production.

Based on the results of this study, we report that the formation of lisogenous aerenchyma in response to root hypoxia stress is regulated by LSD1 and EDS1 and involves at least 37 CWL enzymes in the cell wall saccharification and decomposition process. We also report that conditional PCD regulators (LSD1, EDS1 and PAD4) are developmental regulators in woody plants of the secondary cell wall, since lines with the reduced expression of these genes, growing in four subsequent seasons in the field conditions, in stable transgenic aspen significantly reduced lignin content and significantly increased cellulose fiber polymerization degree. Most importantly, these results are strongly supported by significant similarities (homology) between proteins identified in the MALDI-TOF experiment on Arabidopsis thaliana and DEGs identified in the cambium of transgenic aspen Line 4 (Appendix A).

Usually, the transgenic plants with modified wood composition exhibited impaired growth, especially in the field. Downregulation of cinnamate 4-hydroxylase (an enzyme involved in an early stage of phenylpropanoid synthesis pathway) by RNAi resulted in ~30% less lignin content in hybrid aspen but also reduced tree growth and wood mechanical properties [46]. Another widely used target for genetic manipulation of lignin content is 4-coumarate: coenzyme A ligase (4CL). Suppression of this enzyme leads to a 40–45% reduction in lignin content in aspen [47,48,49], however, this phenotype was confirmed in the greenhouse conditions [50]. During the experiments carried out in the field, the decrease in lignin content was accompanied by stunted growth [51,52,53]. One of the most promising field experiments carried out on poplar with RNAi of 4CL and CCoAOMT resulted in 28% less lignin content and unchanged growth parameters, however, the experiment was performed on 1-year-old plants (one vegetation season) [54]. To sum up, the reduction of lignin is considered often a defense trait against biotic stress. It is not usually studied in the context of plant growth [55,56,57] or reduced-lignin phenotype when growing in the greenhouse. It is also significantly weakened in fluctuating field conditions [17,18,58]. Here, we demonstrate that the deregulation of genes encoding these proteins rather than the lignin biosynthesis pathway itself appears much more promising in biotechnological improvements of trees growing for several subsequent seasons in the field. Most importantly, these regulators control the phenylpropanoid pathway and the genes involved in CWL and lignin biosynthesis (e.g., CAD) in *Arabidopsis thaliana* and aspen (GO:0045551, Appendix A). Moreover, it is shown that the deregulation of the conditional PCD regulators results also in the deregulation of many genes from the CAZY family in the cambium of transgenic aspen Line 4 in the field conditions [59,60,61].

Lowering the lignin level up to 22% in transgenic aspen lines reduced their growth only slightly in comparison to the wild type (Appendix A). Additionally, a higher degree of cellulose polymerization (up to 23%) was found in all transgenic lines. The higher cellulose polymerization facilitates the formation of hydrogen bonds among neighboring cellulose fiber and thus plays a pivotal role in increasing the mechanical performance of wood, easier separation of lignin and hemicellulose from cellulose fibers, simultaneously suppressing the decomposition of cellulose during lignin removal [62]. These features were desirable to produce bioethanol from the wood of trees growing in the field for four subsequent seasons (optimal period for harvesting of fast-growing aspen plantation). Indeed, it was confirmed during experiments on a semi-industrial scale. Considering the above, the deregulation of well-known conditional PCD regulators (*LSD1*, *EDS1* and *PAD4*) [19,44,45,63] gives an unexpected effect in a completely new context i.e., important for the biofuel industry.

Apart from improving wood as a raw technological material, the improvement of the enzymatic hydrolysis and fermentation process could be considered through the identification of the unknown so far plant CWL enzymes. In the industry, the most abundant enzymes are those obtained from the fungus *Aspergillus niger*. Many of the identified *Arabidopsis thaliana* proteins taking part in the lysigenous aerenchyma formation and aspen genes involved in the cell wall modifications have many unknown functions, thus they give us a wide range of opportunities to search for new biotechnological innovations for bioethanol, paper, production process or improvement of wood quality.

## 5. Conclusions

To sum up, we demonstrate for the first time that the deregulation of genes encoding conditional PCD regulators i.e., LSD1, EDS1 and PAD4 can be a breakthrough biotechnology for the biofuel industry. We clearly showed that LSD1 and EDS1 were involved in the plant cell wall saccharification and hypoxia stress and, moreover, we demonstrated that the deregulation in *PtLSD1*, *PtEDS1*, and *PtPAD4* obtained by genetic enginery results in trees are a better feedstock for biofuel production. Although it needs to be further investigated, some of our discoveries can be applied directly to improve bioethanol production. Our hypothetical model of the role of LSD1, EDS1 and PAD4 in conditional regulation of cell death, stress responses and desired utility feature of wood (level of lignin and cellulose polymerization degree) is presented in Figure 4.

## Figures and Tables

**Figure 1 cells-12-02018-f001:**
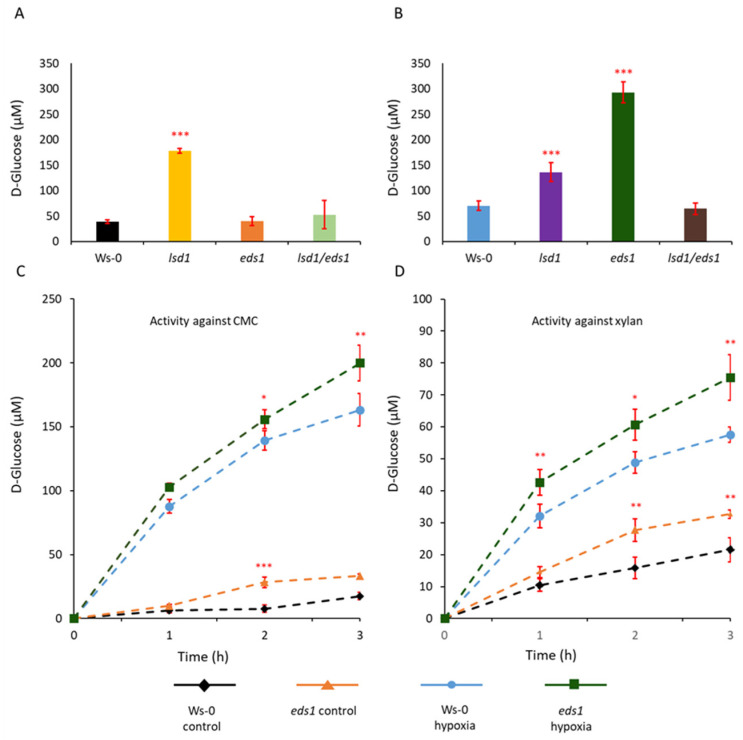
Lysigenous aerenchyma formation in *Arabidopsis thaliana* roots and hypocotyls in response to roots hypoxia stress monitored by reducing sugar content and in vitro activity of cellulolytic and xylanolytic enzymes. The reducing sugar content was measured before (**A**) and after (**B**) 7 days of hypoxia stress expressed as D-glucose content. Analysis of the hydrolytic activity of protein extracts against carboxymethyl cellulose (**C**) and xylan (**D**) was measured in controlled conditions and after 7-day-hypoxia stress which caused the release of reducing sugars during 3-h incubation. Means values (±SD) were calculated from a minimum of three and a maximum of six separate biological replicates (n = 3 to 6). Stars above the bars indicate statistically significant differences in comparison to the Ws-0 plants, according to Tukey HSD at levels *p* < 0.05 (*), *p* < 0.01 (**), and *p* < 0.001 (***).

**Figure 2 cells-12-02018-f002:**
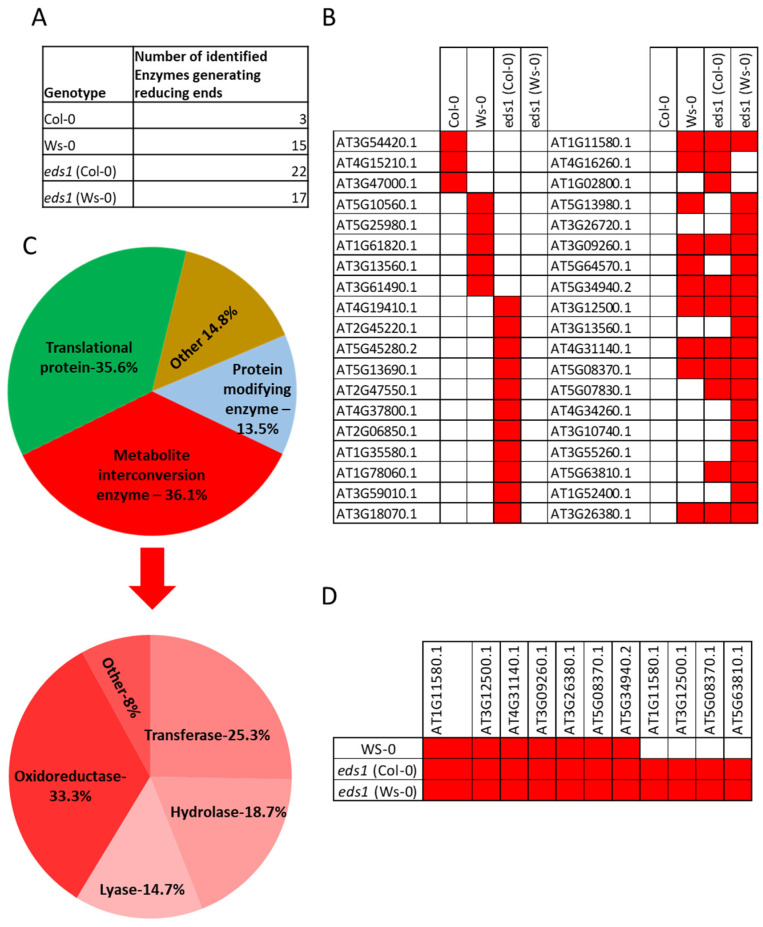
Identification of proteins isolated from roots and hypocotyls during lysogenic aerenchyma formation due to root hypoxia stress by the Maldi-TOF analysis. A number of enzymes generating reducing ends (**A**) and their presence in particular genotypes (**B**). Protein classes analysis by the Panther is shown as a pie chart presenting different protein classes taking part in root hypoxia stress response and their abundance among identified proteins in *eds1* (Col-0) and a pie chart presenting the division of categories metabolite interconversion enzyme (PC00262) into subcategories (**C**). EDS1 abundance dependent generating reducing ends enzymes (**D**) Data are based on six separate experiments (biological replicates) and two technical repetitions *per* sample. The results were analyzed using the Mascot software and trimmed so there will be no more than 1% of false positives in each sample.

**Figure 3 cells-12-02018-f003:**
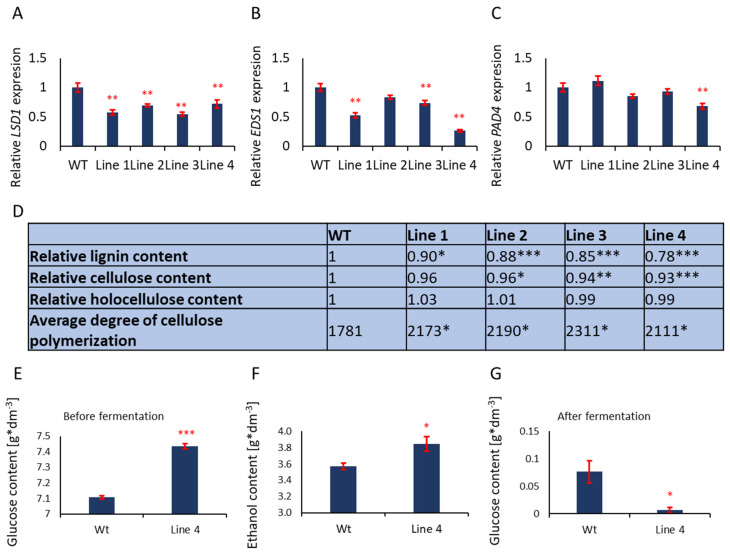
Integrated deregulation of Pt*LSD1*, Pt*EDS1,* and Pt*PAD4* affected lignin content and other wood chemical properties. Transgenic aspen trees (Lines 1–4) grew for four subsequent seasons in field conditions. The relative expression level of *PtLSD1*, *PtEDS1* and *PtPAD4* in leaves (**A**–**C**), relative lignin, cellulose, holocellulose content and a degree of cellulose polymerization (**D**), glucose content after saccharification of wood (**E**), the amount of ethanol in the fermentation process (**F**) and the amount of un-fermented glucose in the pulp after fermentation (**G**). Mean values (±SD) are derived for three different biological and three technical replications for each transgenic Line on graphs (**A**–**C**) (n = 9) and for three to six separate biological replications (n = 3 to 6) for data (**D**–**G**). Stars above the bars indicate statistically significant differences in comparison to the WT plants, according to Tukey HSD at levels *p* < 0.05 (*), *p* < 0.01 (**), and *p* < 0.001 (***).

**Figure 4 cells-12-02018-f004:**
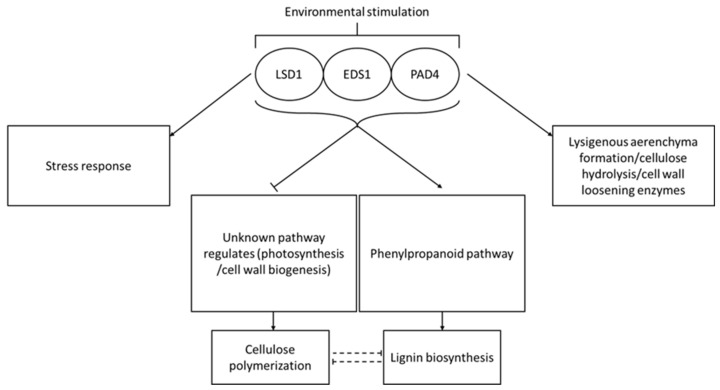
Hypothetical model of the role of LSD1, EDS1 and PAD4 in conditional regulation of the cell wall properties. Based on our results we propose the following model of the role of LSD1/EDS1/PAD4 trimer in conditional regulation of lignin biosynthesis and cellulose polymerization degree during plant growth in a natural environment. Both in Arabidopsis hypocotyls undergoing lisogenous aerenchyma formation proteomic analysis and from xylem/floem transcriptome analysis in transgenic Line 4, many proteins/genes from the phenylpropanoid pathway were deregulated (Appendix A). Knowing that this is one of the most important pathways in the lignin biosynthesis process we proposed this as the main reason for lower lignin content in the cell wall of transgenic aspen lines. The molecular/biochemical pathway of cellulose polymerization degree could be regulated by other unknown regulators (11%) as a response to transgenic deregulation of LSD1, EDS1 and PAD4. We hypothesize that a higher cellulose polymerization degree in transgenic Line 4 is a form of compensation for reduced lignin content.

## Data Availability

All data generated or analyzed during this study are included in the very published article and its Appendix A.

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
