# Peer review of "Biotechnological Potential of the Stress Response and Plant Cell Death Regulators Proteins in the Biofuel Industry"

_cells, 2023, doi:10.3390/cells12162018_

Round 1

Reviewer 1 Report

This is an important work on the relationship between cell wall development and cell death mediated by LSD1 and EDS1 regulators in plants grown under hypoxia conditions. The work also provides beneficial information about a potential application of lsd1- and/or eds1-mutated plants.

Before recommending acceptance of this manuscript, I would ask the authors to consider my comments below.

Abstract

P1L29: “members”  “Members”

Introduction

P2L62-64: Please check the grammar of this sentence again.

P2L75: “Arabidopsis thaliana” should be italic.

Results

P2L97: “both LSD1 and is are” may be “both LSD1 and EDS1 are”.

P2L93-100: The glucose content of the lsd1/eds1 double mutant was comparable to that of the wild-type control both before and after hypoxia, although the glucose content of each single mutant significantly increased before and/or after hypoxia (Fig. 1 A and B). The authors attempted to interpret the results, but their discussion is not clear. I would ask the authors to explain the reason why the glucose content of the double mutant did not increase more clearly.  

P3L121-124: The proteins from the eds1 mutant showed the higher hydrolysis activity against polysaccharides compared with those from the wild-type control (Fig. 1 C and B). The authors described this is probably because the proteins from the eds1 mutant are unaffected by endogenous proteolysis. If it so, why the proteins from the lsd1/eds1 double mutant did not show a higher hydrolysis activity against polysaccharides compared with those from the wild-type control (Fig. S2)?

P6L194: Lignin is an inhibitor of enzymatic polysaccharide saccharification as well as alcohol fermentation. Did the saccharification efficiency of polysaccharides (cellulose and xylans) change in the transgenic aspen line #4? Lignocellulose from the transgenic line contained the higher amount of glucose than that from the wild-type control before enzymatic saccharification probably because the runaway cell death occurred in the transgenic line. This possibly increased the ethanol productivity. However, the authors did not mention about the enzymatic saccharification efficiency of polysaccharides, which may be affected by lignin content. 

P7L226-229: A growth penalty is a critical problem for application of decreased and/or modified lignin plants, but it also suggests physiological significance of lignin in plants. According to Fig. S10 C and D, the stem growth was apparently inhibited in the transgenic lines compared with the wild-type control. Is not there statistical significance in the difference? 

Author Response

Dear Reviewer,

On behalf of myself and all the contributors, I would like to thank you very much for your insightful review. We have followed your remarks and comments, and we would like to submit to you an improved version of the manuscript titled "Biotechnological potential of the stress response and plant cell death regulators proteins in the biofuel industry". We attach our manuscript in docx file in track changes mode.

As requested, we expanded the introduction, described why we used two different species and how they relate to each other. In addition, the manuscript was once again thoroughly checked for spelling and grammatical errors.

Rev 1 comment - P1L29: “members” → “Members”

                It is corrected.

Rev 1 comment - P2L62-64: Please check the grammar of this sentence again.

                We thoroughly checked this sentence and corrected it so that it is fully grammatically correct.

Rev 1 comment - P2L75: “Arabidopsis thaliana” should be italic.

                The font is changed to italic.

Rev 1 comment - P2L97: “both LSD1 and is are” may be “both LSD1 and EDS1 are”.

                It is corrected.

Rev 1 comment - P2L93-100: The glucose content of the lsd1/eds1 double mutant was comparable to that of the wild-type control both before and after hypoxia, although the glucose content of each single mutant significantly increased before and/or after hypoxia (Fig. 1 A and B). The authors attempted to interpret the results, but their discussion is not clear. I would ask the authors to explain the reason why the glucose content of the double mutant did not increase more clearly. 

We have heeded your comment and explained Figure 2 in more detail. You will find this in the paragraph titled “EDS1 and LSD1 are involved in lysigenous aerenchyma formation.”

Rev 1 comment - The proteins from the eds1 mutant showed the higher hydrolysis activity against polysaccharides compared with those from the wild-type control (Fig. 1 C and B). The authors described this is probably because the proteins from the eds1 mutant are unaffected by endogenous proteolysis. If it so, why the proteins from the lsd1/eds1 double mutant did not show a higher hydrolysis activity against polysaccharides compared with those from the wild-type control (Fig. S2)?

This is again related to the LSD1-EDS1 interaction and their antagonistic relationship. These proteins mutually regulate each other, and when a plant lacks both functional LSD1 and EDS1, its phenotype is usually close to the wild-type. We showed this relationship in many of our publications also cited in this manuscript. [e.g. 15, 16]

Rev 1 comment - Lignin is an inhibitor of enzymatic polysaccharide saccharification as well as alcohol fermentation. Did the saccharification efficiency of polysaccharides (cellulose and xylans) change in the transgenic aspen line #4? Lignocellulose from the transgenic line contained the higher amount of glucose than that from the wild-type control before enzymatic saccharification probably because the runaway cell death occurred in the transgenic line. This possibly increased the ethanol productivity. However, the authors did not mention about the enzymatic saccharification efficiency of polysaccharides, which may be affected by lignin content.

Figure 3 shows the glucose level after enzymatic hydrolysis performed in both lines (WT and Line 4). We did not check the content of monosaccharides in the plant material before treatment with enzymes or alkaline pretreatment, therefore, we cannot say what effect runaway cell death had here on the glucose yield. In Line 4, we obtained more glucose after alkaline pre-treatment and enzymatic hydrolysis and fermentation of pulp from line 4 yielded in more ethanol. Therefore, we can conclude that the lower content of lignin which is an inhibitor in both processes, played a key role here.

Rev 1 comment - A growth penalty is a critical problem for application of decreased and/or modified lignin plants, but it also suggests physiological significance of lignin in plants. According to Fig. S10 C and D, the stem growth was apparently inhibited in the transgenic lines compared with the wild-type control. Is not there statistical significance in the difference?

Yes, a main shoot diameter and average annual growth in trunk thickness were lower in the transgenic line than in the wild type, but they were not statistically significant. However, it did not affect the weight of individual trees (average for WT - 8kg and average Line 4 - 7.3kg) (Fig S10 B ), (no statistically significant ) and it is the weight of the raw material that is crucial in the production process.

Reviewer 2 Report

The manuscript of Bernacki et al. describes the functional connection between cell death regulatory genes - LESION SIMULATING DISEASE 1 (LSD1), 31 PHYTOALEXIN DEFICIENT 4 (PAD4) and ENHANCED DISEASE SUSCEPTIBILITY 1 (EDS1) on cell wall composition in roots of Arabidopsis thaliana and wood of aspen (poplar). The authors employed genetically modified plants in both cases and used a battery of diverse techniques, including enzymatic activity assay, MALDI-TOF, RNASeq or fermentation process analyses. Overall, this is a nicely written paper of acceptable quality, however its major weakness is that the experimental part shows some shortcomings (see below). Consequently, the present version of the manuscript is, in my opinion, insufficient and should be developed.

The major points of criticism are:

1)      The majority of the paper bases on the big data and cell wall composition analysis of transgenic Arabidopsis and poplar plants. However, the authors show absolutely no evidence for the transgene expression in the analyzed lines, beside the expected phenotypic effects (knock-down of gene expression).  These experiments should begin from multiple controls indicating the genetic background of used plants (transgene expression analysis in case of poplar and PCR evidence for homozygous lines of Arabidopsis mutants) in order to eliminate any possibility of stress-induced changes in the expression of analyzed genes. This aspect is crucial, as the vast majority of the paper builds on these plants.

2)      The introductory part of the paper does not introduce the reader adequately why these two experimental model plants were used for the study? It is quite confusing that there is no clear explanation and experimental shift from Arabidopsis to poplar. I suggest to rewrite the introductory part in order to help the potential readers to follow the content.

3)      The Material and Method section devoted to Cellulase and xylanase activity assays is not sufficiently explained and in consequence makes impossible for the readers to repeat this experiments in their own laboratory. This should be revised.  

In general, this could potentially be of interest to the scientific community studying plant cell wall polysaccharides and their applicative potential for biofuels production. However, because of some shortcomings of the content, this paper in its present form is still not complete and should be developed.

English language is relatively fine. No major issues detected.

Round 2

Reviewer 2 Report

The authors responded to all my comments. I have no further commnets to the corrected version of the manuscript.